# Non-obstructive intracellular nanolasers

Alasdair H. Fikouras [1], Marcel Schubert [1], Markus Karl [1], Jothi D. Kumar[1], Simon J. Powis [2], Andrea Di Falco[1] & Malte C. Gather [1]

Molecular dyes, plasmonic nanoparticles and colloidal quantum dots are widely used in biomedical optics. Their operation is usually governed by spontaneous processes, which results in broad spectral features and limited signal-to-noise ratio, thus restricting opportunities for spectral multiplexing and sensing. Lasers provide the ultimate spectral definition and background suppression, and their integration with cells has recently been demonstrated. However, laser size and threshold remain problematic. Here, we report on the design, high-throughput fabrication and intracellular integration of semiconductor nanodisk lasers. By exploiting the large optical gain and high refractive index of GaInP/AlGaInP quantum wells, we obtain lasers with volumes 1000-fold smaller than the eukaryotic nucleus ($V_{laser} < 0.1 \, \mu m^3$), lasing thresholds 500-fold below the pulse energies typically used in two-photon microscopy ($E_{th} \approx 0.13 \, pJ$), and excellent spectral stability (<50 pm wavelength shift). Multiplexed labeling with these lasers allows cell-tracking through micro-pores, thus providing a powerful tool to study cell migration and cancer invasion.

[1] SUPA, School of Physics and Astronomy, University of St Andrews, North Haugh, St Andrews KY16 9SS Scotland, UK. [2] School of Medicine, University of St Andrews, North Haugh, St Andrews KY16 9TF Scotland, UK. Correspondence and requests for materials should be addressed to A.D.F. (email: adf10@st-andrews.ac.uk) or to M.C.G. (email: mcg6@st-andrews.ac.uk)

In an effort to complement nanophotonic objects that interact with light via spontaneous processes[1–7], lasers embedded in tissue or even in the cytoplasm of a cell have recently been used for high-density optical barcoding of cells and to perform local optical sensing, and have been suggested as non-linear probe for super-resolution imaging[8–15]. The lasers used so far occupy a substantial fraction of the cell volume (typical volume of eukaryotes, 1000–10,000 $\mu m^3$; volume of the nucleus, the largest stiff component in most eukaryotes, $\approx 100 \mu m^3$). Lasing within cells has been achieved from the whispering gallery mode (WGM)[16] resonances supported by fluorescent polystyrene spheres and oil droplets with diameters >10 $\mu m$ (volume, >500 $\mu m^3$)[9,10]. Very recently, intracellular lasing from nanowires with linear rather than spherical form-factor has been demonstrated, with reduced volume ($\leq 1 \mu m^3$) but with lengths in the range of 3–8 $\mu m$[11,12]. However, for many studies one will require sub-$\mu m$ size in each dimension, e.g. to allow migration of cells through capillaries (typical diameter, 5 $\mu m$) and epithelial layers (pore and channel size in migration and nuclear rupture assays, 1.5 $\mu m$)[17].

Miniaturization of lasers, down to deep sub-wavelength dimensions, is an area of very active research in optical computing and communication and is widely regarded as one of the most promising avenues to address the ever-increasing demand for speed and bandwidth in data transmission and information processing[18–20]. However, the performance targets and trade-offs pertaining to miniaturization of intracellular lasers are considerably different: intracellular lasers operate in an aqueous environment which poses demands on their chemical stability and leads to reduced refractive index contrast between the laser material and its environment; the laser material, the resonator, and the pump should have the least possible impact on cell physiology; to integrate intracellular lasers with other bioimaging technology platforms, they should operate in a spectral window already used for in vivo microscopy and where tissue scattering and absorption are low.

The intracellular lasers developed here are formed by WGM disk resonators with minute volumes ($\sim 0.1 \mu m^3$) and sub-$\mu m$ diameters (down to 700 nm). The laser material is an epitaxially grown aluminum gallium indium phosphide (AlGaInP) multi-quantum-well structure that provides large optical gain[21,22]. This material platform has the additional benefit of being free of the highly toxic arsenic often found in other III–V semiconductor lasers[23,24]. We found that our disk lasers are readily internalized by a variety of cells, including e.g. by human T-cells, which are too small to engulf previously reported polymer-, oil- or nanowire-based intracellular lasers. The WGMs supported by each disk have low loss and showed lasing upon optical pumping of the disk with sub-pico-joule pulses of light. We show that the lasing wavelength is highly dependent on disk size, allowing unique labeling of large numbers of cells by introducing one or several nanodisk lasers of distinct diameter. Importantly, the nanodisk lasers introduced here are small enough to allow unobstructed migration of cells in confined environments, which is demonstrated using NIH 3T3 cells as an example.

## Results

**Design and fabrication of nanodisk lasers.** In practice, the miniaturization of most laser resonators is hampered by the sharp increase in losses at small sizes; efficient lasers typically require resonators with $Q$ factors above $10^3$–$10^4$. Figure 1a shows a schematic illustration of a cell that has internalized one of the WGM nanodisk lasers developed here. For WGM resonators like these, the overall $Q$-factor is given by a radiative component, $Q_{rad}$, as well as a number of other components, including surface scattering and material absorption. $Q_{rad}$ strongly depends on the

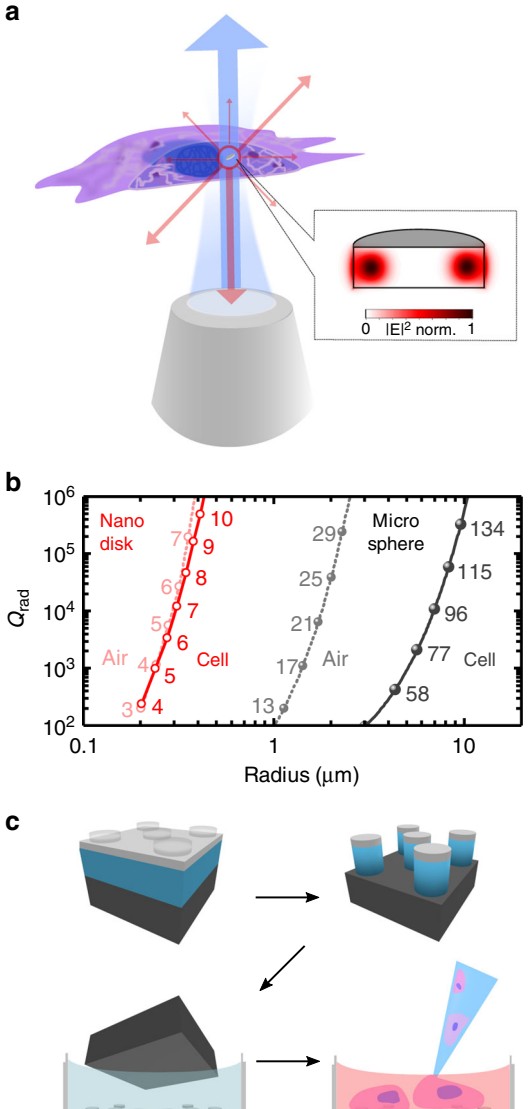

**Fig. 1** Concept and modeling of intracellular nanodisk lasers. **a** Illustration of a semiconductor nanodisk laser internalized into a cell. The disk is optically pumped through a microscope objective (blue) with laser emission (red) collected by the same objective. Insert shows the calculated profile of the lowest radial order transverse electric (TE) mode for a 750 nm diameter disk made of a GaInP/AlGaInP quantum-well structure. **b** Finite element modeling of the radiative $Q$-factor of lowest radial order TE modes in whispering gallery mode micro-resonators with different radii. Comparison between GaInP/AlGaInP nanodisks and polystyrene microspheres, both placed either in air or within a cell. Numbers next to each symbol indicate the angular quantum number of the corresponding mode; the vacuum wavelength for all modes was kept fixed at 680 nm. Lines are guides to the eye. **c** Schematic illustration of the nanolithography based fabrication of nanodisk lasers, the under-etch process for transfer into cell nutrient medium, and the subsequent culture of cells for disk internalization

resonator diameter and on the refractive index contrast with the environment and hence is of particular importance here (Fig. 1b, Supplementary Figs. 1 and 2). Previously reported intracellular lasers were based on microsphere WGM resonators made from materials with $n_{sphere} \approx 1.6$. Optical modeling shows that the low index contrast between these spheres and cells (typical $n_{cell} \approx 1.37$)[25] requires a minimum sphere diameter of 10 $\mu m$ to reach $Q = 10^3$. By contrast, our AlGaInP quantum-well structures

provide an average refractive index of $n_{AlGaInP} \approx 3.6$ and we therefore expect to achieve sufficient $Q$ for lasing in cells with resonator diameters well below 1 µm.

A further advantage of producing intracellular lasers from a semiconductor quantum-well structure epitaxially grown onto a support wafer is the availability of well-established electron beam-based nanolithography processes to accurately control size and shape. To render these processes compatible with biological samples, we developed a combined under-etch and washing process to detach the nanodisks from their support wafer and transfer them into a cell culture medium (Fig. 1c, Methods).

**Laser characterization outside cells.** Figure 2a shows a scanning electron microscopy picture of our nanodisk lasers. In this example, the disk detachment process was interrupted just before completing the under-etch, leaving individual disks with diameters around 750 nm that rest on slender pillars. Using this configuration, we established that if there is a gradual minute change in disk size across the wafer, the lasing wavelength can be tuned continuously (Fig. 2b), in line with predictions by optical modeling (Supplementary Figs. 3 and 4). This variation in wavelength between lasers is useful for optical barcoding of cells as illustrated below.

To confirm lasing action and characterize the laser performance, disks were transferred into a Petri dish filled with cell culture medium. Individual disks were excited by a pulsed diode-pumped solid-state laser that was coupled to an optical microscope (pump wavelength, 473 nm; pulse duration, 1.5 ns; diameter of pump spot on sample, 4 µm). The emission from the disk was collected through the same objective, filtered from the pump light and passed to a digital camera and a spectrometer (spectral resolution, $\approx 0.2$ nm FWHM). Figure 2c shows the intensity of light emitted by our nanodisks versus the peak intensity of the pump pulses, revealing a characteristic s-shape behavior on the log-log plot, with a lasing threshold fluence of $E_{th}$ = 30 µJ cm$^{-2}$ (i.e., lasing started if ≥0.13 pJ pulse energy was incident on a nanodisk). Around the lasing threshold, a sharp spectral line emerged on top of the initially broad photoluminescence spectrum of the nanodisk and at higher pump intensities, the emission spectrum was dominated entirely by the lasing peak (signal-to-background ratio, 24 dB; FWHM of peak, 0.2 nm, i.e. limited by the resolution of the spectrometer). Our nanodisk lasers showed laser emission at <100 µJ cm$^{-2}$ pump fluence for 12 weeks when kept under physiological conditions (i.e., in cell culture medium and at 37 °C). During 50 min of continuous laser operation under these conditions (i.e., >$10^5$ pump pulses, the longest we tested so far), the peak lasing wavelength fluctuated by less than 50 pm (Fig. 2e, determined by peak fitting to the lasing spectra, $N = 3$).

**Internalization of nanodisk lasers.** A range of different cell types readily internalized our nanodisk lasers (Fig. 3a, Supplementary Figs. 5 and 6, Supplementary Movie 1), presumably via natural endocytosis or phagocytosis[26]. For NIH 3T3 cells, we observed a disk uptake efficiency of $(76 \pm 5)$% (Supplementary Fig. 6). Importantly, internalization was also observed for T cells; approximately 20% of investigated cells contained at least one disk $(N = 80)$. T cells are too small to engulf the previously reported cell lasers and represent an important target for cell tagging due to their complex role in cancer progression and immunotherapy[23,27]. Uptake of nanodisk lasers was also observed into the soma of cultured primary neurons, which is typically not substantially larger than the cell nucleus it contains, again illustrating the importance of using sub-µm sized lasers for cell internalization. The presence of the nanodisk laser had no

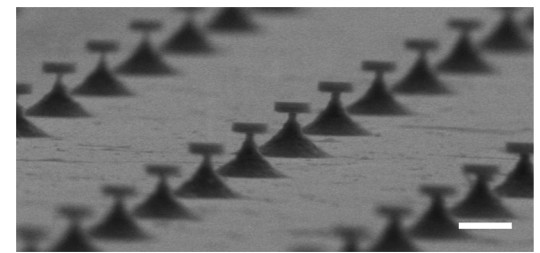

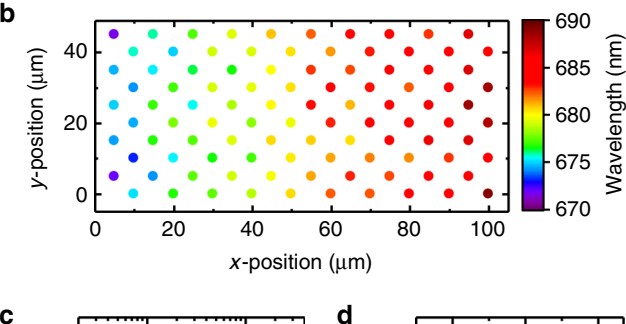

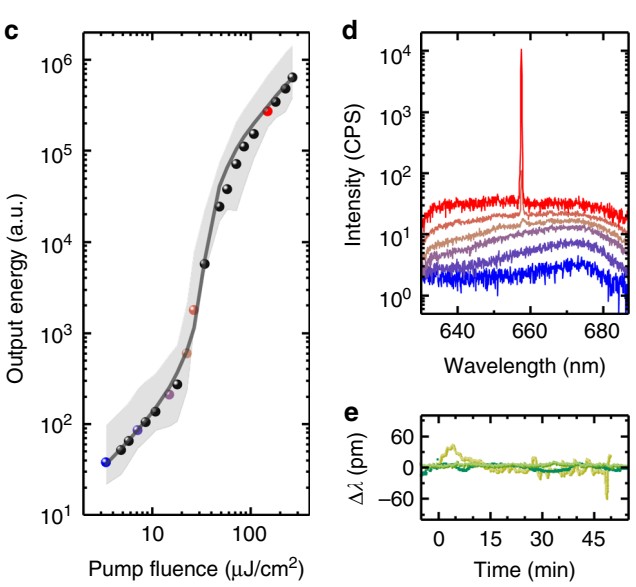

**Fig. 2** Nanodisk laser characterization. **a** Scanning electron microscopy image of an array of as fabricated nanodisks on struts. Scale bar, 1 µm. **b** Color map illustrating the lasing wavelengths for a region of 100 nanodisks on the sample from (**a**), showing an increase in wavelength with increasing disk diameter. **c** Log-log plot of light intensity emitted by detached nanodisk lasers in cell medium as a function of pump intensity. Average threshold curve for $N = 10$ nanodisks (symbols), min/max output energy band of same disks (gray band) and fit to the average with rate-equation model (gray line). **d** Emission spectra for a representative nanodisk at pump fluences corresponding with the colored symbols in (**c**). **e**, Spectral stability of lasing peak for three typical detached nanodisks in cell medium under continuous excitation at 100 Hz; intensity of pump pulses, 100 µJ cm$^{-2}$

noticeable effect on cell behavior and cell viability. Specifically, cell viability/cell death assays performed on NIH 3T3 cells showed no significant difference in the viability of cells in disk-containing samples versus disk-free controls after four days of incubation (Supplementary Fig. 7). In addition, we found that the proliferation rate of NIH 3T3 cells was not affected by the presence and internalization of nanodisk lasers (doubling time, $\tau_{dbl,disk} = (22.65 \pm 0.95)$ h versus $\tau_{dbl,control} = (22.55 \pm 1.93)$ h,

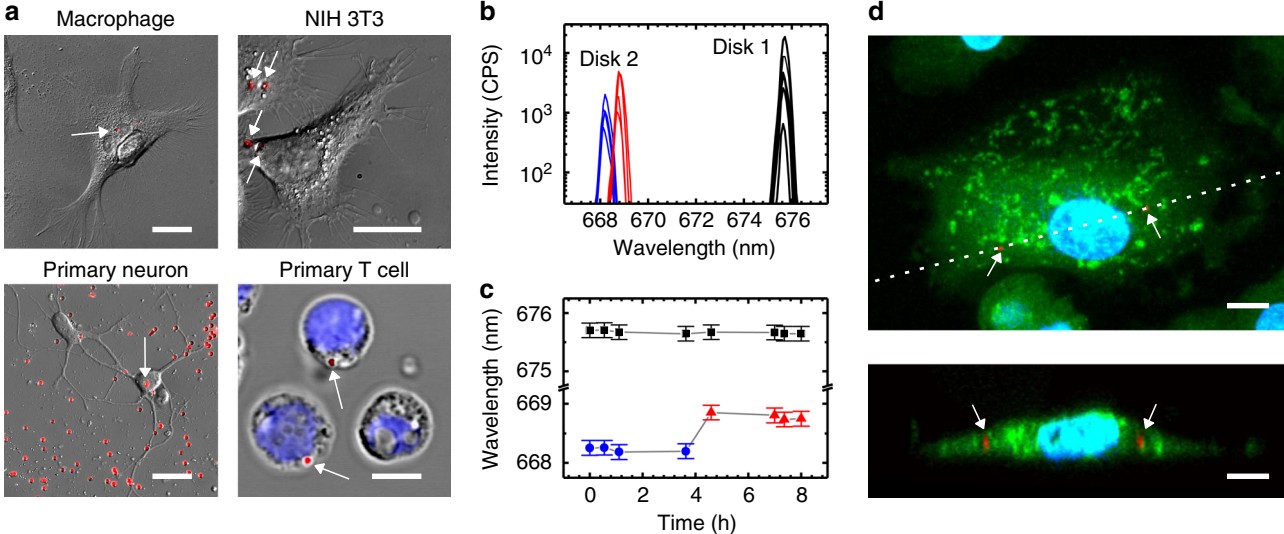

**Fig. 3** Cellular uptake and lasing from semiconductor nanodisks. **a** Differential interference contrast (DIC) microscopy of primary human macrophage, NIH 3T3s, primary mouse neurons and primary human T cells with internalized nanodisks (overlaid red fluorescence, indicated by white arrows). Nucleus of T cells labeled by blue Hoechst dye. **b** Laser spectra collected over period of 8 h for nanodisk inside a macrophage (Disk 1, black) and of a second disk that is internalized by the same cell during the experiment (Disk 2; blue before uptake, red after uptake). **c** Peak wavelength for spectra in (**b**) over time. Error bars indicate FWHM of spectra. **d** Laser scanning confocal fluorescence microscopy image of fixed macrophage with nanodisks (red fluorescence, indicated by white arrows), nucleus in blue (Hoechst), and cytosol in green (Calcein-AM). Maximum intensity projection (top) and vertical cross-section along the dotted line in the top panel. All scale bars, 20 μm; except for primary T cell, 5 μm

Supplementary Fig. 8). Cultures of primary cells containing disks were found to remain viable for at least 2 weeks.

Nanodisks continued to produce narrowband laser light when inside a cell (Fig. 3b; FWHM, 0.2–0.25 nm; over the course of this study, lasing spectra from intracellular nanodisks were observed for >20 independent cell cultures). We followed a nanodisk laser inside a macrophage cell over 8 h and found that its intensity varied over time, presumably because cellular motion induces changes in the position and orientation of the nanodisk, thus affecting the amount of emission that is collected by the microscope objective. These fluctuations prevent accurate recording of a light-in/light-out curve in live cells. As the Q-factor of our high refractive index nanodisks is expected to reduce only marginally when going from cell culture medium ($n = 1.33$) into the cell ($n_{cell} \approx 1.37$), we expect that disk internalization by cells has a minimal effect on their threshold. This assumption is supported by the fact that the lasing threshold of a disk located in a fixed cell was not significantly different from the threshold of disks in cell culture medium (Supplementary Fig. 9). In addition, nanodisks located in live cells reliably showed lasing well above threshold at pump fluences $\leq 100$ μJ cm$^{-2}$. Importantly, the lasing wavelength of the intracellular nanodisk remained constant over time, i.e. the peak lasing wavelength changed by <80 pm (Fig. 3c). Lasing spectra also maintained high signal-to-background ratios of above 20 dB. During this experiment, the cell internalized a further nanodisk laser and the change in the refractive index from cell culture medium to inside the cell caused a clearly resolvable shift in lasing wavelength of 0.6 nm, consistent with optical modeling results (Supplementary Fig. 4). Confocal microscopy performed on the cells that were fixed and stained at the end of the lasing experiment confirmed complete cellular internalization of the nanodisk lasers (Fig. 3d).

**Optical barcoding and laser tagging in confined space**. Due to their small size, each cell can internalize multiple nanodisk lasers without adding a heavy payload. The multiplexed emission spectrum of these lasers provides a highly characteristic optical barcode, thus allowing to uniquely label large numbers of cells. Figure 4 shows lasing spectra and microscopy images of NIH 3T3 cells with $N = 1, 2, 3$, and 6 lasers, and compares spectra recorded 1 h apart. The spectral shape of all lasers remained highly conserved. The emission from the $N = 6$ cell illustrates that even very closely spaced lasing peaks can be resolved separately due to the large signal-to-background ratio of the laser emission. As a conservative estimate, we assume that lasers can be clearly identified if their peak wavelengths are >0.4 nm apart (i.e. five-fold more than the maximum wavelength shift observed here). Given the wavelength tuning range of our nanodisk lasers (40 nm) and our ability to tune lasing wavelength during fabrication (Fig. 2b) and assuming there are $N = 6$ disks per cell, one could thus uniquely tag >$10^9$ cells. (Initial tests indicate that it is possible to label cells with $N \geq 6$ disks, Supplementary Fig. 6, although further optimization may be necessary.)

Finally, we demonstrate that cells tagged with nanodisks can migrate within spatially confined environments. For this, we adapted a transwell migration assay that is widely used in cell biology, e.g. to study tumor cell invasion, transepithelial transport and paracrine cell-cell interactions (Fig. 5a). A nutrient gradient was employed as driving force to trigger migration of eGFP expressing NIH 3T3 cells through the pores of the transwell membrane. By integrating a confocal microscope with the laser characterization setup, we were able to observe the top and the bottom side of the membrane. Multiple laser tagged cells were found to cross the membrane. Figure 5b shows confocal microscopy data of a region with one disk-containing cell that has migrated through the membrane and one cell that has not. (In this experiment, >10 cells with disks were found to have crossed the membrane in the investigated 1 mm × 1 mm area; membrane crossing of disk-containing cells was observed in four independent experiments.) Nanodisk lasers on either side of the membrane emitted characteristic, narrowband lasing spectra (Fig. 5c). This demonstrates how these lasers can be used to barcode individual cells during migration through spatially confined environments.

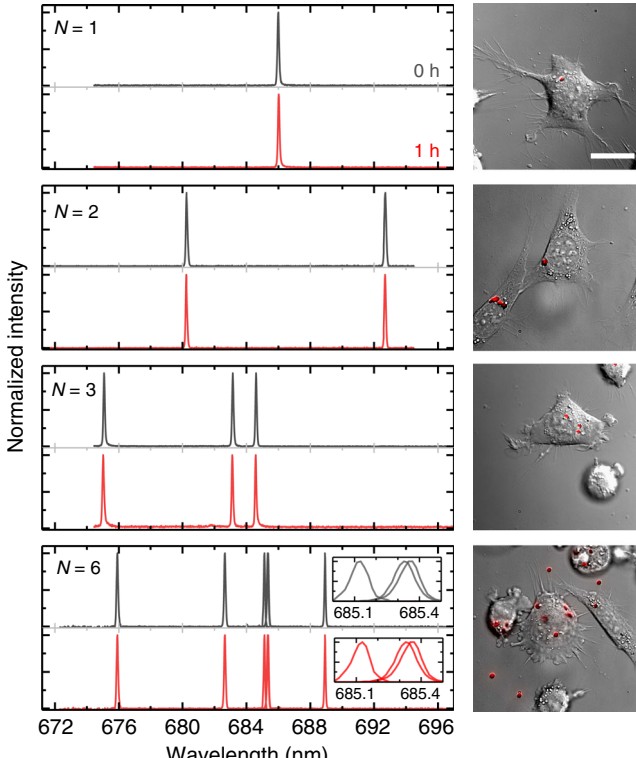

**Fig. 4** Demonstration of optical barcoding of cells with multiple nanodisk lasers. Emission spectrum from NIH 3T3 cells with $N = 1, 2, 3$, and 6 internalized nanodisk lasers, comparing spectra at the beginning of the experiment and after 1 h (left). DIC microscopy images of the same cells with overlaid red fluorescence from nanodisks (right). Inset for $N = 6$ shows spectra collected for individual excitation of three lasers with similar emission spectra. Scale bar, 20 μm

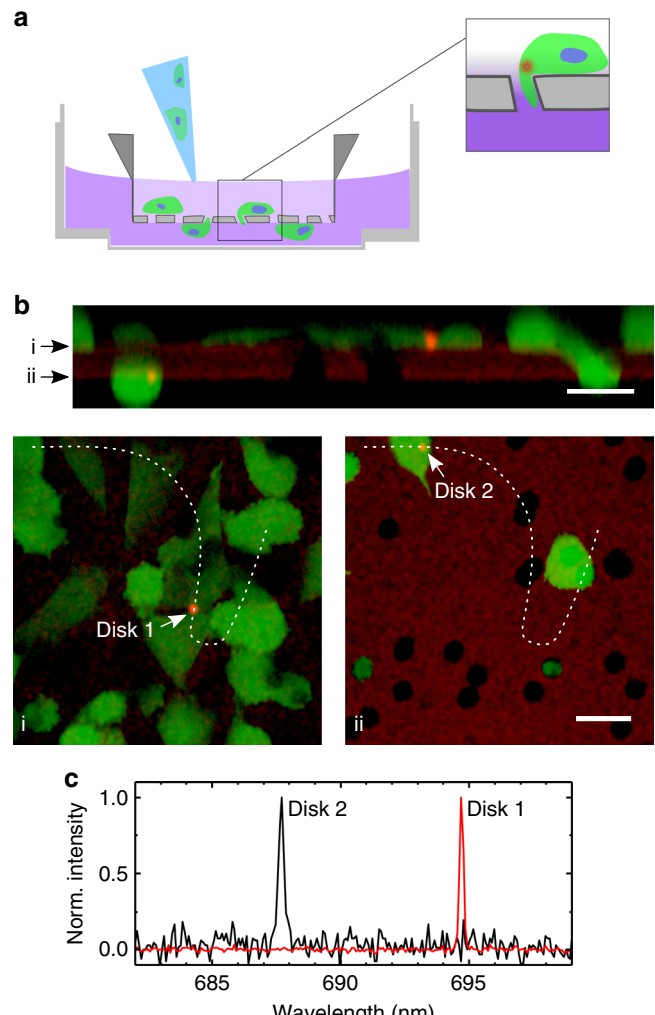

**Fig. 5** Migration of cells with nanodisk lasers through a microporous membrane. **a** Schematic cross-section of transwell migration assay, with cell culture dish, membrane insert and disk-containing cells that are seeded into the low-nutrient medium on the top side of the membrane before migrating through pores in membrane towards nutrient-rich medium on the lower side. **b** Live cell laser scanning confocal microscopy of membrane insert (weak red auto-fluorescence) with eGFP labeled NIH 3T3 cells (green) and nanodisk lasers (bright red). Arrows in top panel indicate the xy-slices shown in the bottom panels; dashed lines in bottom panels indicate the path of the cross-section in the top panel. Images use a logarithmic color scale to visualize weak membrane fluorescence and bright disk fluorescence simultaneously. Scale bars, 20 μm. **c** Lasing spectra of Disks 1 and 2 in (**b**), recorded in parallel with confocal microscopy

## Discussion

The combination of low lasing threshold and bright narrowband emission within the detection range of silicon-based cameras and detectors makes the semiconductor nanodisk lasers introduced here an attractive candidate for a range of biological assays. Due to their small size and compact form factor, they are particularly attractive for applications requiring non-obstructive tagging of cells. Cell migration through pores and epithelial layers plays a crucial role, e.g. in cancer invasion and immune response[28]. Many cells migrate through pores with sizes down to ≈ 5 μm and under certain conditions cells unravel nuclear DNA to pass through openings as small as 1.5 μm[17,29]. Using our lasers to unambiguously track the trajectory of individual cells during these processes will provide important new insights, particularly when combined with a single cell-specific version of fluorescence activated cell sorting[30] as well as with single-cell genomics and proteomics[31].

## Methods

**Nanodisk fabrication.** Nanodisks were produced from an epitaxially grown quantum-well structure located on a sacrificial AlGaAs layer and a sacrificial GaAs wafer (EPSRC National Centre for III-V Technologies, Sheffield). The specific layer structure used was GaAs (substrate), $Al_{0.77}Ga_{0.23}As$ (800 nm, sacrificial), InGaP (10 nm, protection), $(Al_{0.7}Ga_{0.3})_{0.51}In_{0.49}P$ (58 nm), $2 \times [(Al_{0.5}Ga_{0.5})_{0.51}In_{0.49}P$ (10 nm), $Ga_{0.41}In_{0.59}P$ (7 nm)], $(Al_{0.5}Ga_{0.5})_{0.51}In_{0.49}P$ (10 nm), $(Al_{0.7}Ga_{0.3})_{0.51}In_{0.49}P$ (58 nm), InGaP (10 nm, protection). A 500 nm-thick layer of SU8 2000.5 resist (Microchem) was spun onto the substrate, patterned by electron beam lithography at 30 kV (Raith E-line plus), and crosslinked on a hot plate at 90 °C for 2 min. After development in Ethyl Lactate for 1 min, the sample was cured for 20 min at 180 °C, and exposed to low-pressure oxygen plasma for 1 min to fully clear the unexposed substrate. Samples were then immediately etched in a 2:5:100 $Br_2$:HBr:$H_2O$ solution for 10 s to define columns. Following complete removal of SU8 in oxygen

plasma, samples were placed upside down in a plastic cell culture dish (Ibidi μ-dish, 35 mm, #81156) with 2.5% HF:$H_2O$ solution to detach disks directly into the dish. To eliminate residuals of the etchants before introducing cells, the content of the culture dish was neutralized with concentration and volume matched NaOH and then washed with DI water and phosphate buffered saline (PBS) to remove the soluble NaF byproduct. Finally, dishes were filled with culture medium. In the present study, typically 5–10 million disks were produced in each fabrication batch.

**Cell culture and microscopy.** NIH 3T3 cells were purchased from Sigma Aldrich and used at passage numbers <15. For isolation of human macrophages and T cells, peripheral blood was obtained from normal healthy donors after ethical review (School of Medicine, University of St Andrews, no. MD10814) and written informed consent. Cells were isolated from blood following the same protocol as described in ref. [9].

Primary mouse neurons were obtained from C57 mouse pups (sacrificed postnatal day 2–3). Animal procedures were in accordance with the United Kingdom Animals (Scientific Procedures) Act 1986 and were approved by the

University of St Andrews Animal Welfare and Ethics Committee. Hippocampi from unsexed pups were dissected and pooled in ice-cold DPBS. The tissue was treated with papain (10 units/ml) for 22 min at 37 °C, and subsequently disaggregated by repeated pipetting in DMEM/F-12 containing 10% FBS. Cells were pelleted and resuspended in Neurobasal-A supplemented with B-27 and 0.5 mM GlutaMax-I (Thermo Fisher Scientific).

For lasing experiments, cells were cultured in DMEM or $CO_2$ independent RPMI with 10 vol% fetal bovine serum (FBS, Thermo Fisher Scientific) and 1 vol% penicillin-streptomycin (PS). To introduce nanodisk lasers into cells, cells were trypsinised using trypsin-EDTA (0.25%, Thermo Fisher Scientific) and seeded into disk-containing culture dishes and incubated as usual. Cells were maintained in a humidified incubator at 37 °C and 5% $CO_2$.

For the cell proliferation assay a nanodisk containing sample was prepared by detaching approximately 100,000 nanolasers into a 35 mm petri dish, following the procedure described above. A second sample that contained no nanolasers and that was not exposed to the nanolaser detachment procedure served as control. Approximately 5000 NIH 3T3 cells $cm^{-2}$ were seeded in a 35 mm petri dish using DMEM (phenol red free) substituted with 10% FBS, 1% Glutamax, 1% non-essential amino acids (NEAA), and 1% PS, and kept in a heated incubator at 37 °C and 5% $CO_2$ for over 4 days (cell medium replaced every 2 days). Using a 20x objective, cells were imaged every 24 h on an inverted microscope (Nikon TE2000) equipped with differential interference contrast (DIC) optics.

For the cell viability assay, a nanodisk containing sample and a control sample was prepared as described above. 35 mm petri dishes were seeded with 5000 NIH 3T3 cells/$cm^2$ and incubated for 4 days in DMEM (phenol red free) substituted with 10% FBS, 1% Glutamax, 1% non-essential amino acids (NEAA), and 1% PS. The cell medium was replaced after 2 days. Cells were stained for 15 min with Hoechst 33342 (10 nM, Calbiochem, Cat# 382065), Calcein-AM (2 μM, Calbiochem, Cat# 206700) and Propidium Iodide (5 nM, Calbiochem, Cat# 537059) in HBSS. The staining solution was removed, the cells were washed once with PBS, fixed for 10 min in 4% paraformaldehyde, and washed once more with PBS. Gentle and slow pipetting was used during all washing steps to avoid removal of dead cells or nanolasers. Cell viability was assessed by the green fluorescence of Calcein-AM (activated by cytosolic esterase in live cells) and by red fluorescence from Propidium Iodide (a red-fluorescent cell-impermeable nucleic acid stain that stains dead cells with ruptured membranes). The total number of cells were obtained by automatically counting Hoechst-labeled nuclei. Dead cells were identified by colocalization of Hoechst and Propidium Iodide.

3D confocal imaging (Fig. 3 and Supplementary Fig. 6) was performed on a Leica TCS SP8 laser scanning microscope with 20×, 40×, and 63× oil immersion objectives. The same staining and fixation protocol as described for the dead/live assay was used. Hoechst, Calcein-AM, Propidium Iodide, and nanolasers were excited by sequentially scanned continuous wave lasers with a wavelength of 405 nm, 488 nm, 561 nm, and 633 nm, respectively.

Transwell migration assays were performed using inserts with translucent PET membranes with 8 μm diameter pores (Greiner Bio-One). Inserts were mounted in coverslip-bottom Petri dishes (Ibidi) using a custom holder that allowed in situ confocal sectioning of cells and disks with a high NA oil immersion objective. The top side of the membrane insert was seeded with eGFP expressing NIH 3T3 cells containing nanodisks in FBS depleted medium (containing 0.1 vol% FBS). A nutrient gradient was established by filling the bottom compartment with FBS rich medium (30 vol%). Prior to imaging, cells were incubated overnight under the culture conditions described above (i.e., humidified incubator at 37 °C and 5% $CO_2$). Confocal imaging was performed on a Nikon TE2000 equipped with a C1si confocal scanhead, using a 60x oil immersion objective. The sample was excited with a 488 nm Ar ion laser and green and red fluorescence from eGFP and nanodisks, respectively, was detected simultaneously through separate photomultiplier tubes with corresponding filter blocks.

**Optical setup.** Nanodisk containing cells were characterized on an inverted optical microscope, equipped with epi fluorescence, differential interference contrast (DIC) and a laser scanning confocal scanhead (Nikon TE2000 and C1si). The output from a pulsed diode-pumped solid-state laser (Alphalas), which was set to 100 Hz repetition rate, was coupled into the microscope via a dichroic filter and passed to the sample through either a 60× oil immersion or a 40× long working distance objective. The pump fluence was adjusted with neutral density filters and unless stated otherwise a fluence of ≈ 100 μJ $cm^{-2}$ was incident on the nanodisks for recording the lasing spectrum. Emission from disks was collected by the same objective, separated from the pump light by the dichroic and passed to the camera port of the microscope. The image of the sample was then relayed to a 300 mm spectrograph (Andor) and a cooled sCMOS camera (Hamamatsu). A green band pass filter placed in the dia illumination path of the microscope, a removable dichroic beam splitter, and additional band pass filters at the spectrograph and camera allowed simultaneous recording of the spectral output of nanodisks and DIC imaging of cells. The setup also allowed combining investigation of lasers with recording of epi fluorescence and confocal microscopy images (see transwell migration assay above), without need to move the sample. A motorized xy stage with trigger interface (Prior Scientific) enabled automated sequential excitation of a large number of disks across an extended area of the sample. Long-term live cell imaging was rendered possible through the use of an on-stage incubator system (Bioscience Tools).

**Modeling.** Radiative $Q$ factors and resonant wavelengths of disks with different radii and in different media were obtained via finite element modeling (COMSOL Multiphysics), using the perfectly matched layer (PML) approach to obtain the complex eigenfrequencies for modes within the optical gain region[32,33]. PML thickness, z-distance, offset, and growth factor were optimized to avoid numerical instabilities. Disks were modeled as solid isotropic structures with a uniform, isotropic refractive index of 3.6. Radiative $Q$ factors of spheres were modeled using a semi-classical (WKB) approximation for the Riccati-Bessel radial solutions[16].

**Reporting Summary.** Further information on research design is available in the Nature Research Reporting Summary linked to this article.

## Data availability
The datasets supporting this publication can be accessed via the PURE repository at https://doi.org/10.17630/998faf40-3b86-466f-af09-288dd106606c.

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

## Acknowledgements

We thank Liam O'Faolain (CIT, Ireland) for fruitful initial discussion, Andrew Morton for support with neuronal culture, and Gareth Miles for kind provision of neuronal tissue samples. This research was financially supported by the European Research Council under the European Union's Horizon 2020 Framework Programme (FP/2014-2020)/ERC Grant Agreement No. 640012 (ABLASE), by EPSRC (EP/P030017/1, EP/L017008/1) and by the RS Macdonald Charitable Trust. AHF and MK acknowledge support through the EPSRC DTP (EP/M508214/1, EP/M506631/1). MS acknowledges funding by the European Commission (Marie Sklodowska-Curie Individual Fellowship, 659213) and the Royal Society (Dorothy Hodgkin Fellowship, DH160102).

## Author contributions

A.H.F. fabricated the nanodisks. A.H.F. and M.S. characterized nanodisks and carried out the cell experiments. M.K. performed optical modelling. J.D.K. optimized the transwell migration assay. S.J.P. was responsible for cultures of primary macrophages and T cells. A.D.F. and M.C.G. supervised the project. M.C.G. wrote the manuscript with input from all authors.

## Additional information

**Competing interests:** The authors declare no competing interests.

