## [Peer Review File · Nature Communications]

Reviewers' comments:

Reviewer #1 (Remarks to the Author):

This paper looks great - clear on motivation and high on innovation. On the assumption that cell tracking is an important application, then the need for these small lasers is cleared based on the Q-factor arguments leading to lasers made from more conventional materials being much larger than the ones demonstrated here. This, then, is the way to do such studies, and the demonstration across a range of cell types and cell sizes is key. The authors have shown particle differentiation and tracking during cell migration. I think for a first demonstration paper this is all that should be expected.

As such I think that the work is certainly significant enough for Nature Communications and recommend publication.

I have have only two small comments that could be considered further:

1. The authors note the semiconductor materials used aren't as toxic as they could be, but is there any evidence that such materials might be toxic to any cell type?
2. I'd assume that the discs change their orientation during the experiments and that this would then impact the orientation of the emission and the output energy. I'm unclear if this is what the authors are seeing when they talk the intensity changing due to 'cellular motion' on line 109? This should be clarified. Is there any data on how the energy changes with orientation?

Reviewer #2 (Remarks to the Author):

The present work has three primary claims:

- Development of a novel nanodisk imaging agent based on AlGaInP. By changing the index contrast, low threshold emission is possible.
- Demonstration of imaging in several different cell types based on phagocytosis with no cytotoxicity to the cells
- Demonstration of bio-barcoding and cell migration tracking

The authors present concrete evidence supporting the first claim. However, the second and third claims are missing key pieces of experimental data. Additionally, for all data and results presented, the manuscript is lacking sufficient detail in experimental methods as well as statistical analysis regarding the reproducibility of the results. Therefore, I am unable to support the publication. More detailed feedback is below.

Major concerns:

Does wavelength of emission follow expected theoretical prediction based on excitation source and environment? Does the change in emission as the laser disk moves from cell media through the high index cell membrane and into the cell follow an expected shift?

It is proposed that the nanodisk transport relies on endocytosis, which is an active transport process that requires ATP. Where is the ATP source?

Typically, endocytosis is verified using TEM cross sections, not confocal, due to the resolution required to image the vacuole. An alternative approach is to perform FRET with an internalized fluorophore. The presented data is not at a subsequently high resolution to verify the precise location within the cell (internalized within the cell vs. localized in the membrane). Therefore, the claims of endocytosis (or phagocytosis) are not experimentally supported.

Similar to the previous comment, one of the claims in the manuscript is bio-barcoding. In order to

barcode, it is necessary to be able to precisely label cells with well-defined wavelengths. Among many parameters, this requires two things to be combined: 1) surface chemistry to target specific cells and 2) imaging agents with well-defined spectra. Neither of these key metrics have been shown in this manuscript. More importantly, no control over the uptake (or absorption) is shown in this work. Therefore, the claim of bio-barcoding is pre-mature at this time.

One claim by the authors, "Importantly, internalization was also observed for T cells which are too small to engulf the previously reported cell lasers and which represent an important target for cell tagging due to their complex role in cancer progression and immunotherapy. Uptake of nanodisk lasers was also observed into the soma of cultured primary neurons, which is typically not substantially larger than the cell nucleus it contains, again illustrating the importance of using sub- μm sized lasers for cell internalization." As mentioned previously, the data supporting this statement is unclear.

One claim by the authors, "The presence of the nanodisk laser had no noticeable effect on cell behaviour and cell viability. Specifically, a cell viability / cell death assay showed 100% viability of disk-containing macrophages after 24 h ($n = 110$ cells); cultures of primary cells containing disks remained viable for at least 2 weeks." No data is shown to support this statement. More importantly, it would be extremely surprising to see these results for any cell types (regardless of treatment). Cells typically have to be split according to a regular protocol, and recording precisely 100% cell viability is unheard of in biological measurements. Error in biological measurements is normally 5-15%. To be clear, to make statements regarding "cell behaviour and cell viability", live/dead assays, cell counting, and signaling protein production assays (of relevant proteins for the specific cell line) would need to be performed. These should be compared to control cells (not exposed to the nanodisks). All data should be presented, and appropriate statistical analysis should be performed. The duration should be determined by the replication rate of the cell type.

Most importantly, no statistical analysis or reporting of number of trials is shown. At a minimum, this should include: 1) FWHM of lasers, 2) previously mentioned biostatistical analysis (plotted over time by cell type), and 3) threshold (in cell media and in cell reported by cell type).

Minor concerns:

Figure 1, which is mentioned near the beginning of the paper, is located in the methods section, which is at the end. From this reviewer's perspective, this is strange as it requires the reader to flip to the end. However, this is a personal opinion.

The Q radiation expression is not entirely correct. It overlooks several terms, including the wavelength. Additionally, there are several other loss mechanisms that are important when designing a resonant cavity including the material loss and surface scattering loss (which, presumably, is why you chose e-beam lithography) as well as coupling loss (which motivates your optical system design). A more rigorous calculation, including modeling as well as other Q terms, as well as discussion regarding your system design should be included in a supplement.

The imaging and cell preparation sections are missing significant experimental details. For example, "standard culture conditions" do not exist. Additionally, given that primary cells were used, information regarding the purification methods should also be included. Confocal imaging conditions are completely missing. All details needed to reproduce results need to be included. Below are some examples of imaging papers with details, in case this request is not clear.

- Quantitative single-molecule imaging by confocal laser scanning microscopy
Vladana Vukojević, Marcus Heidkamp, Yu Ming, Björn Johansson, Lars Terenius, and Rudolf Rigler, PNAS 105 (47) 18176-18181 (2008)
<https://doi.org/10.1073/pnas.0809250105>

- Visualizing hippocampal neurons with in vivo two-photon microscopy using a 1030 nm picosecond pulse laser, Ryosuke Kawakami, Kazuaki Sawada, Aya Sato, Terumasa Hibi, Yuichi Kozawa, Shunichi Sato, Hiroyuki Yokoyama & Tomomi Nemoto, Scientific Reports volume3, Article number: 1014 (2013)

<https://www.nature.com/articles/srep01014>

- Real-time high dynamic range laser scanning microscopy, C. Vinegoni, C. Leon Swisher, P. Fumene Feruglio, R. J. Giedt, D. L. Rousso, S. Stapleton & R. Weissleder, Nature Communications volume7, Article number: 11077 (2016)

<https://www.nature.com/articles/ncomms11077>

Similar to the previous comment, how were cells stored and split? For measurements that are multiple weeks in duration, it is typical to split cells to ensure that nutrients remain constant.

Similar to the previous comment, were the slides for imaging rinsed after preparation? There appear to be numerous non-incorporated nanodisks. This is a concern as it is unclear if the disks are truly incorporated or simply stuck to the cells.

All approval #'s for animal and human trials need to be provided.

As stated by the authors, the fabrication method is poorly controlled. This can be a strength (multiple lasers can be made in parallel) or a weakness (poor control over emission). Have you tried purifying the subsequent solution to have well-defined emission wavelengths?

Similar to the previous comment, in a solution phase synthesis, it is possible to synthesize millions of imaging particles with a narrow size distribution in a single batch. How many nanodisks can be synthesized at once? What is the efficiency of each step?

Reviewer #3 (Remarks to the Author):

This novel work gives a succinct description of how sub-wavelength scale semiconductor disks can be used as narrow band labels for biological applications. I thoroughly enjoyed reading this interesting piece of work and believe it both demonstrates a significant advance on the current state of the art and could go on to enable important applications in the life and medical sciences.

Whilst on the whole I feel the manuscript is appropriate and of sufficient quality to be published in Nature Communications, I have a few concerns which I list below. Should those concerns be sufficiently well addressed I would certainly recommend publication. (Concerns not listed in order of importance.)

General:

For a study working with cells there is very little evidence for repeat measurements. There are essentially no uncertainties shown, probably as a result of the lack of repeats. This is ok to an extent in that this work is about establishing that this s/c disk laser approach works at all, but it would be good to be able to get a better feel for how repeatable the effects described are. Are we being presented with the rare examples of things working out or are these results representative of what happens in the majority of cells? If the latter, why no repeats? If the former then the authors need to be more up front about the challenges remaining.

Specific:

54-55: Say uptake by T-cells, but doesn't say anything about efficiency. How many cells uptake?

(small) Number given later for viability tests, but nothing for efficiency.

76-80: Control of disk size in fabrication? Repeatability from one fabrication run to the next? Homogeneity across wafer? Controlled release, separation etc to produce monodisperse sample of disks or always polydisperse?

Fig. 2d: Need a legend for different spectra in the plot. The range given in c is logarithmic and it is not clear if these spectra are for a linear or log spacing of pump fluences. i.e. how quickly does the narrow emission peak appear on top of the photoluminescence spectrum?

92-94: though the wavelength drift for 50 mins is given, the statement that the disks had stable performance over a 12 week period isn't accompanied by the expected drift value. Is it similar? Much greater? Any dependence on temperature or other environmental variables?

96: what is the uncertainty in the drift? Was this just a one off measurement or were repeat measurements made?

109-110: again, is this just a one off experiment? The drift in the wavelength might be better stated in pm/hr or similar with accompanying uncertainty.

116-125: is the concept of barcoding the primary goal? If so it would seem to make the homogeneity in fabrication less important, or even undesirable. Or is a more deterministic approach wanted to cell labelling? The authors give a value of $>10^9$ unique bar codes. Is this assuming that all cells uptake 6 disks or assuming that some won't? This again comes back to the efficiency/probability of disk incorporation. If most cells will have less than 6 then the number of codes will be reduced accordingly.

131-132: what does "multiple" actually mean here? How many cells crossed the membrane? How many cells didn't? Was there a control experiment to actually show if the presence of the disks made migration more or less likely?

135-143: the authors are a little vague on what the applications might be for the new disk laser tags. One of the co-authors is based in the school of medicine. Does he have an opinion on where these tags could be particularly helpful and a specific motivation for joining the team? Could they be used in conjunction with FACS? If so one could imagine them being used to sample the in vivo distribution of cells in studies of metastasis in small animal studies. Ambitious speculation might not be welcome, but what are the important challenges this approach might help address in the future?

Reviewers' comments:

Reviewer #1 (Remarks to the Author):

This paper looks great - clear on motivation and high on innovation. On the assumption that cell tracking is an important application, then the need for these small lasers is cleared based on the Q-factor arguments leading to lasers made from more conventional materials being much larger than the ones demonstrated here. This, then, is the way to do such studies, and the demonstration across a range of cell types and cell sizes is key. The authors have shown particle differentiation and tracking during cell migration. I think for a first demonstration paper this is all that should be expected.

As such I think that the work is certainly significant enough for Nature Communications and recommend publication.

We thank Reviewer #1 for this very positive assessment of our work!

I have have only two small comments that could be considered further:

1.1. The authors note the semiconductor materials used aren't as toxic as they could be, but is there any evidence that such materials might be toxic to any cell type?

Arsenide is present in many III-V semiconductors used for lasing applications – but not in our lasers. To support our claim of a specifically high toxicity of arsenide and a generally lower toxicity of our laser materials, we would like to point the attention to the following publications:

[1] "Ecotoxicity assessment of ionic As(III), As(V), In(III) and Ga(III) species potentially released from novel III-V semiconductor materials", Zeng C et. al., Ecotoxicology and Environmental Safety (2017), 140, pp: 30-36.

[2] "Acute and chronic arsenic toxicity". Ratnaike R, Postgraduate Medical Journal (2003), 79, pp. 391.

While reference [2] is a highly cited review on the many effects of arsenic toxicity, reference [1] discusses not only arsenide but also indium and gallium and therefore provides a good overview of the relative toxicity of these elements. While toxic effects are found for all three elements above a certain concentration, the study concludes that "In(III) and Ga(III) were not or only mildly toxic", summarizing further that "the results indicate that the ecotoxicity of In(III) and Ga(III) is much lower than that of the As species tested". Both references have been added to the revised manuscript to allow the reader a fair assessment of this important material property (Refs 23 and 24).

Furthermore, our revised manuscript now also contains additional experimental evidence to show that the presence of our nanodisk lasers is not toxic to cells over the lengths of time we have studied so far. See Supplementary Figs 7 and 8.

1.2. I'd assume that the discs change their orientation during the experiments and that this would then impact the orientation of the emission and the output energy. I'm unclear if this is what the authors are seeing when they talk the intensity changing due to 'cellular motion' on line 109? This should be clarified. Is there any data on how the energy changes with orientation?

We indeed assume that the disks can change orientation with time while inside a cell and that this is one of the reasons for changes in the detected output energy. (In addition to lateral motion that will also affect the collection efficiency.) The cumulative effects of disk orientation, free space

coupling of the pump light and scattering inside the cellular environment impede a systematic study of this at the current stage. We note, however, that while the intensity varies over time, a sufficiently intense signal is obtained at all times and our experiments do not indicate that disk orientation constitutes a significant problem. We agree that this was not clear in the original text and have made this clearer in our revised manuscript.

Reviewer #2 (Remarks to the Author):

The present work has three primary claims:

- Development of a novel nanodisk imaging agent based on AlGaInP. By changing the index contrast, low threshold emission is possible.
- Demonstration of imaging in several different cell types based on phagocytosis with no cytotoxicity to the cells
- Demonstration of bio-barcoding and cell migration tracking

The authors present concrete evidence supporting the first claim. However, the second and third claims are missing key pieces of experimental data. Additionally, for all data and results presented, the manuscript is lacking sufficient detail in experimental methods as well as statistical analysis regarding the reproducibility of the results. Therefore, I am unable to support the publication. More detailed feedback is below.

Major concerns:

2.1

Does wavelength of emission follow expected theoretical prediction based on excitation source and environment? Does the change in emission as the laser disk moves from cell media through the high index cell membrane and into the cell follow an expected shift?

We agree that a comparison to optical modelling is useful and of general interest. The revised manuscript now contains supporting information that shows results of more detailed finite element modelling (see Supplementary Figs 1-4).

We pump our lasers well below saturation and excite them non-resonantly. Therefore, we do not expect the wavelength of emission to depend on the excitation source.

Regarding the influence of environment on wavelength: In our experiments, the refractive index distribution in the environment of the disks is inhomogeneous and its exact distribution is unknown. Before the internalization, the disks are separated by a thin layer of water from the irregular surface of the cover slips. Once internalized into the cell, the environment is again inhomogeneous with different intracellular components having different refractive index. It is however instructive to assume a simplified homogeneous environment before and after internalization. In this case, the simulations predict a change in emission wavelength by 10 nm per refractive index unit change in the medium surrounding the disk. In particular, as shown in Supplementary Fig. 4 of the revised manuscript, the modelling predicts a red-shifted of the laser emission of 0.4 nm, when changing the environment from refractive index $n=1.33$ (typical of the cell medium) to $n=1.37$, which is the accepted average refractive index of a cell [Ref 25 in revised manuscript]. This shift is in good agreement to the experimentally observed red-shift of 0.6 nm.

The thickness of the cell membrane is small (typically, 10 nm) compared to the mode volume of our nanodisk lasers ($0.1 \mu\text{m}^3$). We therefore do not expect to resolve a characteristic "change in emission as the laser disk moves from cell media through the high index cell membrane and into

the cell". Experimentally, no additional transient red-shift was seen during internalization (i.e., as compared to after internalization). As the presence or absence of such an effect is not directly required to support our findings, we believe further investigation of this matter is beyond the scope of this initial manuscript on intracellular nanodisk lasers.

Additionally, as discussed further in the following, our calculations show that the wavelength of emission is highly dependent on disk diameter; for a 750 nm diameter disk, a change in disk diameter by 1 nm leads to a change in emission wavelength of 0.84 nm. This compares to statistical fluctuations in emission wavelength over time of less than 0.1 nm.

2.2

It is proposed that the nanodisk transport relies on endocytosis, which is an active transport process that requires ATP. Where is the ATP source?

The ATP required for this process is produced by the cell, using nutrients supplied in the cell culture medium. The internalization of foreign particles by cells that are kept in culture is widely reported in the literature (see e.g. Champion, J. A. & Mitragotri, S. Proc. Natl. Acad. Sci. 103, 4930–4934 (2006); we added this reference to the revised manuscript; see Ref. 26).

2.3

Typically, endocytosis is verified using TEM cross sections, not confocal, due to the resolution required to image the vacuole. An alternative approach is to perform FRET with an internalized fluorophore. The presented data is not at a subsequently high resolution to verify the precise location within the cell (internalized within the cell vs. localized in the membrane). Therefore, the claims of endocytosis (or phagocytosis) are not experimentally supported.

For the present study, the important fact is that the nanodisks are located fully inside the cell, i.e. enclosed by the cell membrane, not whether this occurs through endocytosis, phagocytosis or another mechanism. Our use of laser scanning confocal microscopy for this purpose is justified by the fact that other literature uses the same method to confirm internalization of particles (e.g. Humar/Yun, Nature Photon 9, 572 (2015)). Additionally, in previous work, we have validated the used imaging modalities against a widely accepted fluorescence labelling assay that uses binding between a cell-impermeable, fluorescent streptavidin conjugate and a biotinylated resonator surface to label any non-internalized resonators (Schubert, M et al, Sci Rep 7, 40877 (2015)). We therefore conclude that in order to confirm complete internalization of disks, the resolution provided by laser scanning confocal microscopy (e.g. Fig. 3d) or by combined epi-fluorescence/DIC imaging (Fig. 3a) is perfectly sufficient.

The revised manuscript includes additional supporting information providing further evidence for disk internalization:

Supplementary Fig. 5 shows the primary neuron from Fig. 3a for three different focus settings. In the plane of the cover slip, the disks surrounding the neuron show bright epi-fluorescence, but the disk inside the neuron is hardly visible in the fluorescence channel. Focusing at an intermediate plane above the cover slip, the disk internalized by the neuron shows bright fluorescence but the surrounding disks are out of focus and thus generate a much reduced epi-fluorescence signal. When focusing to the top of the neuronal soma, no noticeable fluorescence is recorded from any of the disks. This confirms that the disk in question is indeed localized above the plane of the cover slip and also not on top of the neuronal soma and thus inside the cell.

Supplementary Fig. 6b-c contains a further laser scanning confocal dataset for NIH 3T3 cells, showing internalization of nanodisks in NIH 3T3s and for full transparency a rare example of a nanodisk found underneath a cell, presumable just before internalization (Slice III). This data is part of a larger experiment that investigates the efficiency of nanodisk uptake by cells (Supplementary Fig. 6a). Investigating a total of 1219 nanodisks, we find an uptake efficiency of $(76 \pm 5)\%$ (sample mean \pm standard deviation).

Finally, to avoid any misunderstanding, we have removed the claim of endocytosis/phagocytosis from the introduction and the caption to Fig. 1 and have rephrased the description of Fig. 3 to say that the cellular uptake happens “presumably via natural endocytosis or phagocytosis”.

2.4

Similar to the previous comment, one of the claims in the manuscript is bio-barcoding. In order to barcode, it is necessary to be able to precisely label cells with well-defined wavelengths. Among many parameters, this requires two things to be combined: 1) surface chemistry to target specific cells and 2) imaging agents with well-defined spectra. Neither of these key metrics have been shown in this manuscript. More importantly, no control over the uptake (or absorption) is shown in this work. Therefore, the claim of bio-barcoding is pre-mature at this time.

We disagree that the two conditions mentioned by the referee are required for useful barcoding. To obtain useful barcoding, one requires the ability to uniquely label a significant fraction of the cells of interest (numbers will depend on application). There is no need to be able to pre-select the wavelength emitted by the label; one can instead rely on statistical variation to ensure that (with a very high probability) each cell is uniquely labelled. Likewise, while it may be useful to label only specific cells, one could equally well label a random population of cells and then only follow the cells of interest. What will be important, though, is the ability to label a substantial number of cells. Our new Supplementary Fig. 6 and earlier data for much larger lasers that we published in Ref. 13 shows that this condition is met.

The query on uptake efficiency was address in response to the previous comment (2.3).

2.5

One claim by the authors, “Importantly, internalization was also observed for T cells which are too small to engulf the previously reported cell lasers and which represent an important target for cell tagging due to their complex role in cancer progression and immunotherapy. Uptake of nanodisk lasers was also observed into the soma of cultured primary neurons, which is typically not substantially larger than the cell nucleus it contains, again illustrating the importance of using sub- μm sized lasers for cell internalization.” As mentioned previously, the data supporting this statement is unclear.

Please see our replies to 2.3 above and to comment 3.2. by referee #3 which address this point.

2.6

One claim by the authors, “The presence of the nanodisk laser had no noticeable effect on cell behaviour and cell viability. Specifically, a cell viability / cell death assay showed 100% viability of disk-containing macrophages after 24 h ($n = 110$ cells); cultures of primary cells containing disks remained viable for at least 2 weeks.” No data is shown to support this statement. More importantly, it would be extremely surprising to see these results for any cell types (regardless of treatment). Cells typically have to be split according to a regular protocol, and recording precisely 100% cell viability is unheard of in biological measurements. Error in biological measurements is normally 5-15%. To be clear, to make statements regarding “cell behaviour and cell viability”,

live/dead assays, cell counting, and signaling protein production assays (of relevant proteins for the specific cell line) would need to be performed. These should be compared to control cells (not exposed to the nanodisks). All data should be presented, and appropriate statistical analysis should be performed. The duration should be determined by the replication rate of the cell type.

We agree that additional data on the question of cell viability is desirable. To address this point we have performed a new series of experiments and included the corresponding data as supporting information in the revised manuscript. Specifically, we performed two independent and longer term (96h) cell viability / cell death assays for NIH 3T3 cells, comparing the viability of cells in the presence of nanodisks to a control experiment where no disks are present (Supplementary Fig. 7). No statistically significant difference is observed between these samples and we counted 95-99% of viable cells in our cultures.

To obtain further information about the response of cells to the presence of nanodisks, we have also performed a cell proliferation assay for the NIH 3T3 cell line, again comparing against a control without disks (Supplementary Fig. 8). As is clear from the data presented, in all of these experiments, we observe no difference between disk-containing dishes and control dishes.

2.7

Most importantly, no statistical analysis or reporting of number of trials is shown. At a minimum, this should include: 1) FWHM of lasers, 2) previously mentioned biostatistical analysis (plotted over time by cell type), and 3) threshold (in cell media and in cell reported by cell type).

FWHM of lasers: We have not optimized our fabrication process for wavelength uniformity as our proposed application in fact requires a variation in the wavelength between different disks (which we achieve with good control as shown in Fig. 2b). To answer the referee's question, we now also analysed a series of nominally identical lasers; please see response to point 3.3 by referee #3 for further details. It should be noted that the range of FWHM would be smaller for a fabrication procedure aimed at obtaining perfectly uniform disk diameters (and hence lasing wavelengths). As explained above, this was not the case for us.

Biostatistical analysis: As explained in detail in our reply to comment 2.6, our revised manuscript now contains extensive data on experiments exploring a possible (adverse) response of cells to the presence of nanodisks. In brief, for NIH 3T3 cells we have performed a proliferation assay over several days. We have also carried out live/dead assays over an extended period of time for 3T3s.

Threshold: Fig. 2c in the original manuscript already shows light-in/light-out curves for n=10 disks in cell culture medium, illustrating that the variation in threshold pump fluence between disks is +/-15% which is perfectly sufficient for our proposed application. Accurate threshold measurements in cells are not possible with our current instrumentation as the output energy of the laser fluctuates due to cellular movement (also see referee #1, point 1.1.). To circumvent this issue we now performed threshold measurements in a fixed cell and compared to the thresholds of disks in culture medium (Supplementary Fig. 9). These show no appreciable difference in threshold as one would expect, because --due to the high refractive index of our nanodisk-- their Q factor is only marginally decreased when going from cell culture medium ($n \sim 1.33$) into the cell ($n \sim 1.37$). This is also supported by the fact that when inside a cell, our nanodisks reliably show lasing with high lasing to background ratio (> 10-20 dB) at pump fluences $\leq 100 \mu\text{J}/\text{cm}^2$. The lasing experiments summarized in Fig. 3-5 of the manuscript were performed at pump fluences $\leq 100 \mu\text{J}/\text{cm}^2$ and we have found that all lasers operated well above threshold at these fluences.

Statements clarifying this have been added to the main text and the methods section of the revised manuscript.

Minor concerns:

2.8

Figure 1, which is mentioned near the beginning of the paper, is located in the methods section, which is at the end. From this referrer's perspective, this is strange as it requires the reader to flip to the end. However, this is a personal opinion.

We have placed all figures and figure captions at the end of the manuscript in line with what we believe is journal guideline. We trust that if our manuscript is accepted for publication, the typesetters will position figures where they are discussed.

2.9

The Q radiation expression is not entirely correct. It overlooks several terms, including the wavelength.

We are confused by the referee's statement that "the Q radiation expression" is not correct. In the manuscript we do not provide an analytical expression for the quality factor, but extract it from numerical simulations that take into account the wavelength of light. This approach is explained in full detail in the Methods section.

Specifically, to avoid artefacts and fluctuations from wavelength changes between resonators of different size, we have kept the wavelength fixed (at 680nm) and calculated the radiative Q factor for modes with different angular quantum numbers, in each case choosing the resonator size such that the resonance wavelength was 680nm. This yields Q factors for a number of discrete resonator sizes; any lines connecting these discrete points are only "a guide to the eye". While this was explained in the caption to Fig. 1b of the original manuscript, we have now expanded the corresponding statement slightly to make this clearer.

2.10

Additionally, there are several other loss mechanisms that are important when designing a resonant cavity including the material loss and surface scattering loss (which, presumably, is why you chose e-beam lithography) as well as coupling loss (which motivates your optical system design).

As the reviewer correctly states, one of the reasons to select e-beam lithography was indeed to minimize surface roughness. The ability to quickly adapt disk diameter and to readily produce structures with sub- μm dimensions were other considerations in selecting e-beam lithography over UV lithography.

We agree that material and scattering losses contribute to the total Q factor of a resonator, following the well-known expression [$Q_{\text{tot}}^{-1} = Q_{\text{rad}}^{-1} + Q_{\text{material}}^{-1} + Q_{\text{surface}}^{-1} + Q_{\text{coupling}}^{-1} \dots$]. In our analysis of Fig. 1b, we only focus on Q_{rad} because the e-beam lithography yields disks with edge roughness below 10nm. Therefore, for our submicron disks, Q_{rad} is two order of magnitude smaller than the typical Q_{surface} and Q_{material} (see e.g. Ref 21 in the revised version of the manuscript). Furthermore, in our configuration Q_{coupling} is already embedded in Q_{rad} , since we only use open radiation channels (assuming a homogeneous environment) to pump the disks and collect the emission, thus one is authorized to assume $Q_{\text{tot}} \sim Q_{\text{rad}}$.

2.11

A more rigorous calculation, including modeling as well as other Q terms, as well as discussion regarding your system design should be included in a supplement.

As explained above, Q_{rad} has been determined through rigorous state-of-the-art optical modelling (using finite element modelling, in this case). To avoid any misunderstanding, further details on the optical modelling have been included as supplementary information (Supplementary Figs 1-4). Also to avoid misunderstanding, the revised manuscript now also mentions the other terms contributing to the overall Q factor.

The optical modelling presented in the original manuscript and the additional supplementary information that was now added is useful to motivate our choice of lasing platform and perform initial estimates. However, the experimental observation of lasing and the lasing spectra are not at all controversial (and were not disputed by any of the referees), and so modelling is not a requirement to validate any of our findings.

2.12

The imaging and cell preparation sections are missing significant experimental details. For example, "standard culture conditions" do not exist.

By "standard conditions" we referred to the conditions explained in detail at the beginning of the "Cell Culture" section of the Methods. For clarity we have rephrased to "under the culture conditions described above". We have also very substantially expanded other parts of the Methods section to provide detail on cell culture conditions during each of the individual experiments.

2.13

Additionally, given that primary cells were used, information regarding the purification methods should also be included.

We thank the reviewer for pointing out that this information has been missing. We have included a reference to an earlier publication from our group that describes the purification procedure for human macrophages in detail (to avoid unnecessary replication). The protocol used for purification of primary mouse neurons is described in detail in the revised manuscript.

2.14

Confocal imaging conditions are completely missing. All details needed to reproduce results need to be included. Below are some examples of imaging papers with details, in case this request is not clear.

- Quantitative single-molecule imaging by confocal laser scanning microscopy

Vladana Vukojević, Marcus Heidkamp, Yu Ming, Björn Johansson, Lars Terenius, and Rudolf Rigler, PNAS 105 (47) 18176-18181 (2008)

<https://doi.org/10.1073/pnas.0809250105>

- Visualizing hippocampal neurons with in vivo two-photon microscopy using a 1030 nm picosecond pulse laser, Ryosuke Kawakami, Kazuaki Sawada, Aya Sato, Terumasa Hibi, Yuichi Kozawa, Shunichi Sato, Hiroyuki Yokoyama & Tomomi Nemoto, Scientific Reports volume3, Article number: 1014 (2013)

<https://www.nature.com/articles/srep01014>

- Real-time high dynamic range laser scanning microscopy, C. Vinegoni, C. Leon Swisher, P. Fumene

Feruglio, R. J. Giedt, D. L. Rousso, S. Stapleton & R. Weissleder, Nature Communications volume7, Article number: 11077 (2016)

<https://www.nature.com/articles/ncomms11077>

Details on confocal imaging conditions have been added to the revised manuscript.

2.15

Similar to the previous comment, how were cells stored and split? For measurements that are multiple weeks in duration, it is typical to split cells to ensure that nutrients remain constant.

Only one of the cell types used in our work is a continuously dividing cell line (NIH 3T3). The other cells are differentiated primary cells and thus do not undergo regular cell division; splitting these cells is therefore not useful. However, nutrient medium indeed should be replaced to ensure constant nutrient supply. For experiments that lasted several days this has been done and details on this are explained in the methods section of the revised manuscript.

2.16

Similar to the previous comment, were the slides for imaging rinsed after preparation? There appear to be numerous non-incorporated nanodisks. This is a concern as it is unclear if the disks are truly incorporated or simply stuck to the cells.

As explained in response to comment 2.3 above, we have performed extensive confocal microscopy and have found that disks are generally internalized rather than being permanently stuck to the cell surface. As we perform extended live cell imaging, we use petri dishes with thin bottom glasses rather than slides, so rinsing is not adequate. However, as explained in response to the previous comment, we have replaced nutrient medium and this has not led to any loss in internalized disks.

2.16b

All approval #'s for animal and human trials need to be provided.

Our work does not contain any animal trials. Mice were kept in a certified animal house that adheres to all relevant regulations (as stated in our manuscript "Animal procedures were in accordance with the United Kingdom Animals (Scientific Procedures) Act 1986." We have extended this statement to say "and were approved by the University of St Andrews Animal Welfare and Ethics Committee"). Animals were sacrificed by a trained individual prior to the start of the experiment via a "schedule 1 method". This does not constitute "regulated procedures" as defined by the act and therefore did not need a Home Office Licence. Additionally, there is no need to apply for permission or licence to execute experiments on cells or tissue derived from sacrificed animals.

As for the experiments on primary human macrophages, we have extended our original statement on ethical review and informed consent, now indicating that we obtain blood samples by venepuncture from normal healthy donors under ethics permission and written informed consent after project review supplied by the School of Medicine ethics committee and have included the relevant project number (MD10814).

2.17

As stated by the authors, the fabrication method is poorly controlled. This can be a strength (multiple lasers can be made in parallel) or a weakness (poor control over emission). Have you tried purifying the subsequent solution to have well-defined emission wavelengths?

The fabrication method guarantees a perfectly suitable accuracy of the disk size distribution for our application. As explained in the manuscript and in our response to other comments (including 2.7 and 3.3), a fluctuation in wavelength is a desirable feature for our experiments and proposed applications. Therefore, work on further purification is beyond the scope of the present work.

2.18

Similar to the previous comment, in a solution phase synthesis, it is possible to synthesize millions of imaging particles with a narrow size distribution in a single batch. How many nanodisks can be synthesized at once? What is the efficiency of each step?

While it is possible to produce millions of imaging particles via solution phase synthesis, these are generally not lasers, certainly not lasers with comparable characteristics in terms of size, threshold and spectral control.

In our current fabrication process, we usually produce between 5 and 10 million lasers in each batch; this information has been added to the methods section of the revised manuscript as it might well be of general interest. We have not performed a systematic quantification of the efficiency of each step but from our observations, we estimate that the overall process, including transfer of disks from the wafer into culture, has an efficiency well above 10%.

Reviewer #3 (Remarks to the Author):

This novel work gives a succinct description of how sub-wavelength scale semiconductor disks can be used as narrow band labels for biological applications. I thoroughly enjoyed reading this interesting piece of work and believe it both demonstrates a significant advance on the current state of the art and could go on to enable important applications in the life and medical sciences.

Whilst on the whole I feel the manuscript is appropriate and of sufficient quality to be published in Nature Communications, I have a few concerns which I list below. Should those concerns be sufficiently well addressed I would certainly recommend publication. (Concerns not listed in order of importance.)

3.1

General:

For a study working with cells there is very little evidence for repeat measurements. There are essentially no uncertainties shown, probably as a result of the lack of repeats. This is ok to an extent in that this work is about establishing that this s/c disk laser approach works at all, but it would be good to be able to get a better feel for how repeatable the effects described are. Are we being presented with the rare examples of things working out or are these results representative of what happens in the majority of cells? If the latter, why no repeats? If the former then the authors need to be more up front about the challenges remaining.

Over the course of developing our nanolasers and the imaging and laser detection we have observed internalization of nanodisk lasers by cells in over 50 cell cultures. We have produced over 30 batches of nanodisk lasers with lasing readily observed for at least the last 25 batches produced, i.e. after the fabrication process had been optimized. As the focus has been on testing the behaviour or lasers in cells and biocompatibility, we have not performed threshold measurements in each case but in a typical successful batch (including the last 25 batches mentioned above) we reliably obtain clear lasing spectra at a pump fluence of $\leq 100 \mu\text{J}/\text{cm}^2$. Within a batch, lasing thresholds between disks vary by $\pm 15\%$ (this has not yet been systematically

optimized). We expect that stability can be improved by further process optimization if required at any point, but note that for the suggested applications the stability achieved is more than sufficient.

In our revised manuscript, we also present new data on the biocompatibility of our nanodisks, and – while there may be more subtle effects that we have not yet detected – we find no sign of changes in cell viability (Supplementary Fig. 7) and (for proliferating cells) no sign of changes in doubling time / replication rate (Supplementary Fig. 8). In addition, we have now included data on the efficiency with which nanodisks are internalized into cells (Supplementary Fig. 6). This data also represents an independent repeat of disk uptake (in addition to data shown in Fig. 3). The data presented in the main manuscript are in most cases representative examples of phenomena that have been observed multiple times in several experiments (generally ≥ 3). For instance, Fig. 4 shows four representative examples of lasing from intracellular nanodisks; for this cell type we have observed similar spectra for >100 cells (but have not recorded high quality microscopy images in each cases). The data shown in Fig. 5 has been challenging to acquire as the scattering membranes render confocal microscopy difficult (which is unrelated to our nanodisks). We have observed >10 nanodisks on the far side of the membrane (i.e., disks that have transversed the membrane), even though we have only been able to investigate a limited area of the sample due to the above imaging challenges. We have observed migration of cells through a membrane in at least three independent experiments.

In addition to the new data mentioned above, we have also included several statements on reproducibility and repeats in the revised manuscript in order to summarize the statements from the above paragraph.

Specific:

3.2

54-55: Say uptake by T-cells, but doesn't say anything about efficiency. How many cells uptake? (small) Number given later for viability tests, but nothing for efficiency.

We have not quantified the uptake efficiency for T-cells systematically. However, when collecting the data for the images in Fig. 3, we observed 16 T-cells with internalized disks among a total of 80 cells investigated. We have now performed a more systematic investigation of uptake efficiency for NIH 3T3 cells (Supplementary Fig. 6).

3.3

76-80: Control of disk size in fabrication? Repeatability from one fabrication run to the next? Homogeneity across wafer? Controlled release, separation etc to produce monodisperse sample of disks or always polydisperse?

Control of disk size has not been optimized in the present study as our proposed application requires variation in size rather than a fixed size. To answer the referee's question, we now also analysed an example of a series of nominally identical lasers and find that with our current process, which has not been optimized for homogeneity, the lasing wavelength varies by 4 nm (min-to-max) and <1 nm (standard deviation); see histogram below. According to optical modelling (see response to point 2.1 and Supplementary Fig. 3), the observed variation in wavelength corresponds to a variation in disk diameter of 5 nm (min-to-max). We assume that this variation is largely due to the use of a wet etch process. Thus – if required at a later stage – uniformity could be improved by changing to dry etching.

We have not systematically tested or optimized size reproducibility from one run to the next and we believe that doing so is beyond the scope of the present work. From experience we can say that after optimizing the fabrication procedure we can reproducibly produce nanodisks that reliably lase at pump fluences $\leq 100 \mu\text{J}/\text{cm}^2$ and that have sub- μm diameters. We have not studied any additional “controlled release, separation etc” procedures to improve monodispersity of our samples. As we do not foresee that this would be required for the proposed application, we find that this is beyond the scope of this study.

3.4

Fig. 2d: Need a legend for different spectra in the plot. The range given in c is logarithmic and it is not clear if these spectra are for a linear or log spacing of pump fluences. i.e. how quickly does the narrow emission peak appear on top of the photoluminescence spectrum?

Thanks for pointing this out. A colour code is now used in the revised manuscript to connect the threshold curve in Fig. 2c with the spectral information in Fig. 2d to identify the corresponding pump fluence for each spectrum.

3.5

92-94: though the wavelength drift for 50 mins is given, the statement that the disks had stable performance over a 12 week period isn't accompanied by the expected drift value. Is it similar? Much greater? Any dependence on temperature or other environmental variables?

We have not continuously recorded lasing from an individual disk for more than 50min. However, in Fig. 3c we have repeatedly (8 times) measured lasing from an individual intracellular nanodisk over the course of 8h and find a maximum change in laser wavelength of $<80\text{pm}$. The statement about 12 weeks refers to the ability of our nanodisks to lase; we have not traced individual disks over this length of time. The corresponding statement in the manuscript has been modified to avoid confusion.

To maintain viability of cells, an on-stage microscope incubation system that keeps temperature constant at 37°C has been used for all experiments. As in most applications of our nanodisk lasers tight temperature control will likely be required to satisfy biology requirements, we have not yet explored temperature dependence.

3.6

96: what is the uncertainty in the drift? Was this just a one off measurement or were repeat measurements made?

To address the question, we now repeated the wavelength fluctuation measurement twice more. The new data has been added to Fig. 2e. The new data show better stability than the original trace (now fluctuating by approximately $\pm 10\text{pm}$). This is likely the result of our recently further improved experimental setup with better overall stability.

3.7

109-110: again, is this just a one off experiment? The drift in the wavelength might be better stated in pm/hr or similar with accompanying uncertainty.

The data shown in Fig. 4 follows a total of four cells and a total of 12 lasers over the course of 1h. Like in Fig. 3c, we again find that wavelength remains stable to within 80 pm. As these data do not point to any possible issue with wavelength stability, we have not performed any additional long-term tracking experiments that investigate possible changes in the wavelength of intracellular lasers over time but have instead focused on collecting further evidence that lasers do not have a detrimental effect on cell viability and proliferation.

As there is no trend in wavelength change over time but instead a random fluctuation, we are unable to convert the observed fluctuation to the requested drift rate in pm/hr and have instead kept peak-to-peak changes over a defined period of time to quantify laser stability.

3.8

116-125: is the concept of barcoding the primary goal? If so it would seem to make the homogeneity in fabrication less important, or even undesirable. Or is a more deterministic approach wanted to cell labelling? The authors give a value of $>10^9$ unique bar codes. Is this assuming that all cells uptake 6 disks or assuming that some won't? This again comes back to the efficiency/probability of disk incorporation. If most cells will have less than 6 then the number of codes will be reduced accordingly.

Indeed, at present the concept of barcoding is the primary goal. This can be achieved by random fluctuation in resonator size or – as illustrated in Fig. 2b – by a continuous gradient in laser size across the wafer.

The $>10^9$ unique barcodes assume $N \geq 6$ disks per cells; we have reworded this in the revised manuscript for better clarity. The revised manuscript now also contains data on disk uptake efficiency, looking at the efficiency of disk uptake by NIH 3T3 cells (Supplementary Fig. 6a). Investigating a total of 1219 nanodisks, we find an uptake efficiency of $(76 \pm 5)\%$ (sample mean \pm standard deviation). The image shows that many cells indeed contain multiple disks (often ≥ 6). To avoid any misunderstanding, we also added a comment to the statement about 10^9 unique labels to say that while achieving $N \geq 6$ disks per cell appears feasible this may need to be optimized in the future.

3.9

131-132: what does “multiple” actually mean here? How many cells crossed the membrane? How many cells didn't? Was there a control experiment to actually show if the presence of the disks made migration more or less likely?

The revised manuscript now includes additional information on the number of cells that were found to have crossed the membrane within the investigated area and on the number of independent experiments during which this has been observed. In short, in the experiment shown in Fig. 5, >10 cells with disks were found to have crossed the membrane in the investigated $1\text{ mm} \times 1\text{ mm}$ area. We performed a total of four independent membrane crossing experiments and while the quality of the confocal images was mixed, examples of disks that had crossed the membrane

were seen in each case.

Due to the presence of the scattering membrane and because of limited working distance of the used high NA microscope objective that is required for optical sectioning of the membrane, confocal imaging proves challenging (a point that is unrelated to the disk lasers). Therefore, we have been unable to quantify the likeliness of migration through the membrane and hence not yet attempted to compare the behaviour to a control experiment.

3.10

135-143: the authors are a little vague on what the applications might be for the new disk laser tags. One of the co-authors is based in the school of medicine. Does he have an opinion on where these tags could be particularly helpful and a specific motivation for joining the team? Could they be used in conjunction with FACS? If so one could imagine them being used to sample the in vivo distribution of cells in studies of metastasis in small animal studies. Ambitious speculation might not be welcome, but what are the important challenges this approach might help address in the future?

Dr Simon Powis is indeed based in the St Andrews School of Medicine. He is an immunologist and is keen to use the nanodisk laser concept to track immune cells (macrophages, dendritic cells, possibly osteoclast), e.g. during migration through epithelial layers. This possibility is discussed in the final paragraph of our manuscript. As suggested by the reviewer, we wanted to avoid overly ambitious speculation on this matter, and so have kept this discussion very concise.

Using or combining laser tagging with FACS is an interesting possibility. This could be either for a cell-based assay on its own or alternatively could allow to identify and isolate individual cells from tissue samples, after an experiment where these very cells have been studied in vivo (or in a tissue slice) using a combination of our laser tracking and other in vivo microscopy modalities (two photon, light sheet etc). FACS may then be an intermediate step, prior to a single cell genomic or proteomic analysis. Again, we would like to avoid lengthy speculation but have added the possibility of using FACS as part of such experiments to the revised manuscript.

REVIEWERS' COMMENTS:

Reviewer #1 (Remarks to the Author):

My view has not changed after reading through the edited manuscript - the other referees have helped to tighten up the biological aspects of the work, and my minor points have also been addressed. I remain comfortable that this is a very strong piece of work and is now suitable for final publication in Nature Comms.

I've read through all the other referee comments and the responses and I think everything has been addressed to the level needed for this initial proof of concept paper.

Reviewer #3 (Remarks to the Author):

Whilst it is a little disappointing that the difficulties in confocal imaging through the membranes has precluded getting some of the efficiencies requested, on the whole the authors have made a good job of addressing my concerns and I can subsequently recommend publication in it's new revised form